# Freeze-on limits bed strength beneath sliding glaciers

Colin R. Meyer [1], Anthony S. Downey[1] & Alan W. Rempel[1]

Discharge from sliding outlet glaciers controls uncertainty in projections for future sea level. Remarkably, over 90% of glacial area is subject to gravitational driving stresses below 150 kPa (median ~70 kPa). Longstanding explanations that appeal to the shear-thinning rheology of ice tend to overpredict driving stresses and are restricted to areas where ice sheets only deform (roughly 50%). Over the more dynamic portions that slide, driving stresses must be balanced by thermo-mechanical interactions that control basal strength. Here we show that median bed strength is comparable to a threshold effective stress set by ice–liquid surface energy and till pore size. Above this threshold, ice infiltrates sediment to produce basal layers of debris-rich ice, even where net melting takes place. We demonstrate that the narrow range of inferred bed strengths can be explained by the mechanical resistance to sliding where roughness is enhanced by heterogeneous freeze-on.

[1] Department of Earth Sciences, University of Oregon, Eugene, OR 97403-1272, USA. Correspondence and requests for materials should be addressed to C.R.M. (email: colinrmeyer@gmail.com)

In his seminal work on the mechanics of glacier flow, Nye[1] remarked on ice-sounding data from 16 Alpine glaciers that revealed a surprisingly narrow range of driving stresses ($\tau \sim$ 50–150 kPa) in comparison with much higher variability in their widths, depths, and slopes[2]. As recognized by Nye and elaborated upon further by Weertman[3], when combined with considerations of mass conservation, the shear-thinning rheology of ice yields a plausible explanation. The argument can be summarized as follows: the scale of stresses needed to shear ice at a characteristic velocity $[U]$ over a characteristic depth $[d]$ implies that $[\tau] \sim [\eta][U]/[d]$, where the ice viscosity follows Glen's law $[\eta] = 1/(2[A][\tau]^{n-1})$ with temperature-dependent ice softness $[A]$ and $n$ is an empirical exponent. Assuming flow to be close to a steady state so that glacier mass is conserved, this ice motion must evacuate net accumulation $[a]$ over upstream length $[l]$, so $[U][d] \sim [a][l]$. This balance is likely responsible for the correlation observed in Greenland's outlet glaciers[4] between velocity and an inverse power of height above buoyancy (which depends on topographic details, but generally scales with $[d]$). Finally, recognizing that the gravitational driving stress acting down the surface slope $[\sin\alpha]$ with weight density $[\rho_i g]$ leads to $[\tau] \sim [d][\rho_i g][\sin\alpha]$, these considerations lead to

$$\left[\tau_{\mathrm{Nye}}\right] \sim \left(\frac{[a][l][\rho_i g]^2[\sin\alpha]^2}{2[A]}\right)^{1/(n+2)} \sim \left(\frac{[U][d][\rho_i g]^2[\sin\alpha]^2}{2[A]}\right)^{1/(n+2)},$$

(1)

where $n \approx 3$ and $[\rho_i g]^2/[A]$ is approximately constant for temperate glacial ice. Equation (1) predicts that even though the characteristic values for $[a]$, $[l]$, $[U]$, $[d]$, and $[\alpha]$ undergo considerable variations, the effect on $[\tau]$ is relatively weak—depending on the products of $[a][l]$ and $[U][d]$ to the small power of $1/(n+2) \approx 0.2$, and on $[\sin\alpha]$ to the power of $2/(n+2) \approx 0.4$.

The scaling argument just described, however, overpredicts the observed driving stresses across large swaths of Greenland and Antarctica. As shown in Supplementary Note 2, there is a considerable scatter in the correlation between deformational stress $[\tau_{\mathrm{deform}}] = [\eta][U]/[d]$ and driving stress $[\tau_{\mathrm{drive}}] = [d][\rho_i g][\sin\alpha]$. Much of this discrepancy can be attributed to sliding at the bottom of the ice sheet, where ice meets the sediment. Allowing for sliding along this interface reduces the deformation velocity $[U]$, which reduces the estimate for $[\tau_{\mathrm{deform}}]$. For sliding ice masses, the average basal shear stress is constrained to match the average bed strength; any connection with ice rheology is indirect. The glacial contribution to eustatic sea-level change is dominated by sliding, which occurs at appreciable rates only where basal ice nears its melting point. Model results by Pattyn[5] indicate that 55% of the grounded portion of the Antarctic ice sheet is in such a thermal state, and a recent synthesis of estimates (see Table 3 in MacGregor et al.[6]) suggests that melting temperatures prevail along a similar proportion of the bed beneath the Greenland ice sheet. Indeed, Rignot and Mouginot[7] report that basal sliding exceeds internal deformation over more than 50% of the Greenland ice sheet; sliding is also common beneath smaller ice caps and valley glaciers[8,9] and governs ice motion for vast regions of Antarctica[10]. Assessments of geomorphic features generated by major Pleistocene ice sheets (see Supplementary Note 1) suggest that extensive sliding has occurred beneath them as well.

An explanation for the narrow range of bed strengths that resist glacier sliding must involve the thermo-mechanical interactions between the underlying substrate and ice. Like most solids, ice is preferentially wetted by its own melt as phase coexistence is approached[11], and the forces exerted between ice and mineral particles across the resulting premelted liquid films support a portion of the glacial overburden. This effective stress determines the curvature of the ice–liquid interface in pore openings and becomes large enough for the ice to infiltrate the underlying sediments once the threshold is surpassed. At higher effective stresses, variations in basal conditions are expected to lead to heterogeneities in freeze-on thickness[12]. Over a broad range of glaciologically relevant conditions, scaling arguments reveal that the roughness imposed by heterogeneous freeze-on produces sliding resistances that match the observed distribution of driving stresses.

## Results

**Narrow range of average driving stress.** Figure 1 illustrates the distribution of average driving stresses implied by ice surface $z_{\mathrm{surface}}$ and bed $z_{\mathrm{bed}}$ elevation measurements in Antarctica (black) and Greenland (green). The median driving stresses of ~60 and 70 kPa, respectively, for these ice sheets are remarkably modest in comparison with median normal stresses $\sigma_{\mathrm{n}} = \rho_i g(z_{\mathrm{surface}} - z_{\mathrm{bed}})$ of ~20 MPa in each case and maximum normal stresses extending to roughly twice that level. Since the average driving stress and resisting basal stress must be approximately in balance over the large areal scales that the constituent data points characterize, these distributions imply that more than three quarters of the basal area of both Greenland and Antarctica is subject to shear stresses below 100 kPa (1 bar); maps of driving stress for these two ice sheets are included in Supplementary Note 2. The exact boundaries between deformation and sliding-dominated regions are disputed[6]. However, the consensus holds that sliding takes place over sufficiently large areas that the range of basal strengths in those regions alone should be well represented by these comprehensive data sets. Furthermore, the average thicknesses and slopes tabulated in the GlaThiDa data set[8] (blue dotted) suggest that despite being subjected to much lower normal stresses (mean: 280 kPa, max.: 2.5 MPa), basal stresses beneath alpine glaciers fall within a similar range.

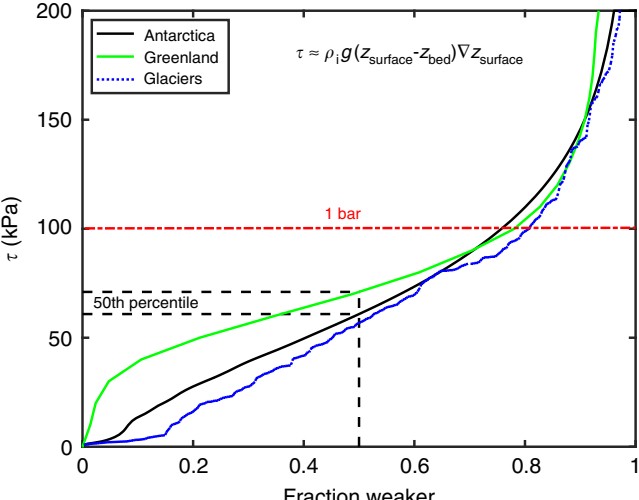

**Fig. 1 Driving stress for glaciers and ice sheets.** Average driving stress for Antarctica (black) and Greenland (green) plotted as a function of the fractional area that is weaker. Antarctic values were calculated from BEDMAP2 data reported on a 1 km grid[30]. Greenland values were extracted from MacGregor et al.[6] based on surface slopes from GIMP[31] and ice thicknesses from Morlighem et al.[27]. Also, shown for comparison (dotted blue) are approximate shear stress values calculated from average thicknesses and slopes (not colocated) plotted as the proportion weaker among 449 pairs of these parameters tabulated for Alpine glaciers in the GlaThiDa data set[8]

Over the vast regions characterized by the narrow range of $\tau$ plotted in Fig. 1, the bed roughness and sliding speed each vary over many orders of magnitude. Bed materials are similarly heterogeneous, but evidence from formerly glaciated landscapes[13] and regions of current fast glacier flow suggests that the discharge from ice sheets is dominated by sliding over beds that are either completely or at least partially mantled by unconsolidated sediments. Even where sliding takes place against exposed bedrock, the ubiquity of striations and moraines in deglaciated terrain and observations of sediment plumes in water bodies adjacent to contemporary ice masses[14,15] testify to the common availability of lithic tools that facilitate the erosive power of glaciers. The load supported by particle contacts in fluid-saturated sediments, referred to as the effective stress $N$, is defined as the difference between normal stress and the pressure supported by pore fluids. In consequence, the frictional component of sliding resistance that can be sourced to clastic contacts is directly proportional to $N$. Since rock friction coefficients $\mu$ are of the order unity, in circumstances where friction dominates, $\tau$ and $N$ are of similar size. These observations suggest pursuing an explanation for the narrow range of $\tau$ beneath sliding ice by analyzing constraints on the range of $N$ under which sliding is permitted.

**Sediment pore-throat curvature induces premelting**. Although some sliding can take place at very cold temperatures[16,17], significant basal motion is limited to conditions that are very close to phase equilibrium. Bulk equilibrium conditions pertain when the free energies of ice and liquid molecules are equal at some particular temperature $T_m$, which is a function of the pressure to which both are subjected (implying that $N = 0$, where $T = T_m$). Two such points are marked with open squares in Fig. 2. Conventional descriptions of ice motion passing small obstacles by regelation rely upon melting and refreezing as a result of the slight depression of $T_m(P_m + \Delta P)$ relative to $T_m(P_m)$ that accompanies a differential pressure of magnitude $\Delta P$ between the upstream and downstream sides. However, as illustrated by an open circle in Fig. 2, equilibrium coexistence between liquid and ice can extend below bulk equilibrium. This situation arises because liquid water preferentially wets ice surfaces to separate them from mineral surfaces, causing the ice pressure to exceed the liquid pressure, with the intermolecular interactions that govern this premelting behavior transmitting stresses between the ice and sediment skeleton to make up the difference[11]. Away from mineral surfaces in the pore throats that separate sediment particles, the difference between the ice and liquid pressures is given by the product of the surface energy $\gamma_{sl} \approx 0.035$ J/m$^2$ with the interfacial curvature. For intuition, in an idealized circular pore throat of radius $r_p$ that separates glacial ice from the liquid-saturated pores beneath, the interfacial curvature can attain a maximum of $2/r_p$; higher values imply that the interface curves so sharply that ice can extend downward and infiltrate the underlying pore space. This sets the threshold pressure difference $p_f$ between the ice and liquid phases when ice first infiltrates basal sediments to $\Delta P = p_f = 2\gamma_{sl}/r_p$ (e.g., $p_f \approx 70$ kPa when $r_p = 1$ μm). More generally, the mean interfacial curvature of the ice–liquid interface at its deepest extent above liquid-saturated pores must attain a uniform value so that $\Delta P = p_f$ when the equilibrium temperature is $T_m - \Delta T \approx T_m[1 - p_f/(\rho_i L)]$, where $\rho_i L$ is the latent heat per unit volume. These considerations imply that sliding of "clean" glacial ice, with $N = \Delta P$, can take place only with $N < p_f$, whereas higher effective stresses are accompanied by ice infiltration into underlying pores and entrainment of ice-bound sediments[12]. Importantly, though particle sizes and hence the characteristic dimensions of the pore throats that separate them

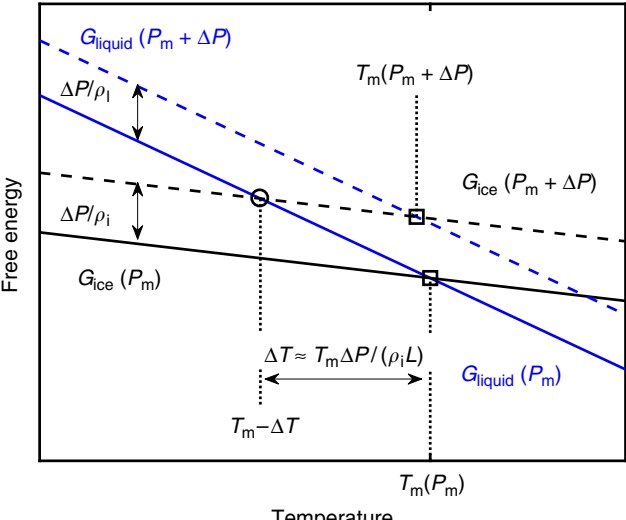

**Fig. 2 Equilibrium coexistence of ice and water.** Illustration of potential equilibrium states implied by changes in Gibbs-free energy satisfying $dG = dP/\rho - sdT$, with $P$ pressure, $\rho$ density, $s$ specific entropy, and $T$ temperature. The steeper (blue) lines depict the entropic reduction to the free energy of liquid water at pressure $P_m$ (solid) and $P_m + \Delta P$ (dashed) as temperature increases. The more shallow (black) lines for ice are somewhat less sensitive to temperature because of the lower-specific entropy of the solid state (i.e., $s_i \approx s_w - L/T_m$, where $L$ is the latent heat of fusion). Conditions where the free energies are equal (open symbols) represent states of thermodynamic equilibrium. The bulk equilibrium temperature $T_m$ at pressure $P_m$ is slightly warmer than that at $P_m + \Delta P$ because the shift in enthalpy with $\Delta P$ for each phase is inversely dependent on its density and $\rho_i < \rho_w$. The surface energy of highly curved ice–liquid interfaces at the entrance to pore throats (and the intermolecular interactions across premelted films) causes the pressure in ice to be higher than that in the liquid so that equilibrium is achieved at $T_m - \Delta T \approx T_m[1 - \Delta P/(\rho_i L)]$ (Dash et al.[11])

vary widely, the mechanical limits on comminution prevent the finest particles from being reduced in diameter below the micron scale[18]; this ensures that locations with $p_f$ exceeding 100 kPa are rare, and is consistent, for example, with observations beneath the Siple Coast ice streams of West Antarctica, where minimum particle sizes are around 1 micron[19].

**Freeze-on thickness as a function of effective pressure**. As $N$ increases towards $p_f$, ice conforms more closely to the geometry of the uppermost clastic surfaces and can drag them over their stationary neighbors while the pore space beneath remains saturated with liquid water. Once $N > p_f$, ice infiltrates into the underlying sediments a distance $h$ that is set by the balance between the gravitational load, the supporting fluid pressure, and the stress transmitted to particle contacts. The characteristic depth $h^\star = \Delta T/G$, for temperature gradient $G$ and $\Delta T = p_f T_m/(\rho_i L)$ provides a useful gauge. Steady-state predictions for the scaled freeze-on thickness $h/h^\star$ as a function of the scaled effective stress $N/p_f$ are illustrated in Fig. 3 (see Methods for details). The central dashed line shows a case in which the liquid pressure distribution throughout the ice-infiltrated zone is hydrostatic; increases in $N/p_f$ are accommodated by an approximately linear increase in $h/h^\star$ so that the temperature at the top of the ice-infiltrated zone cools enough that wetting interactions can transmit the required increased portion of the gravitational load to sediment contacts. The rightmost solid line has ice melting at $\dot{m} = 6$ mm/a (requiring a net heat flux of $\dot{m}/(\rho_i L) \approx 60$ mW/m$^2$), which results in a non-hydrostatic liquid

pressure distribution throughout the ice-infiltrated zone; increases in $N/p_f$ result in comparatively less pronounced increases in $h/h^\star$ because the higher water pressures that drive melt flow offset some of the change in net gravitational load, thereby enabling thinner ice-infiltrated zones since less stress needs to be transmitted to the sediment. The leftmost dot-dashed line shows a case in which net freezing enhances the glacier thickness by 6 mm/a and requires the liquid pressure gradient to steepen so that water can be drawn upward through the ice-infiltrated sediments. With net freezing, more dramatic changes in $h/h^\star$ with $N/p_f$ are expected at first, but a maximum steady-state value of $N/p_f$ is reached at a particular $h/h^\star$ (here ~0.9)—the resistance to liquid transport through thicker ice-infiltrated zones is so great that only lower values of $N/p_f$ can be accommodated. The extension of steady-state curves into the shaded region at larger depths is

unstable[12]. Higher (lower) freezing rates lead to peak stable steady-state values of $h/h^\star$ that are smaller (larger). While no single set of constitutive behavior is able to characterize the broad range of subglacial materials that host sliding, the model curves in Fig. 3 illustrate expected trends. In particular, the ice-infiltrated thickness of basal sediments increases with $N$, decreases with $p_f$ (e.g., accompanying reductions in pore size), decreases with melt rate $\dot{m}$, and decreases with $G$. Moreover, when the heat balance requires net freezing to take place, no steady-state $h$ is stable beyond a small multiple of $h^\star$, as delineated here by the shaded region; instead, transient freezing can continue to large depths and effectively anchor the overlying glacier to its frozen bed, leading to motion by internal ice deformation alone.

**Scaling for bed strength with ice infiltration.** Heterogeneous basal conditions can produce variations in $h$ that alter the topography of the basal slip surface. Though the geometrical perturbations along any particular flow line will be dictated by the local hydrology through $N$, the geology through $p_f$, and the heat transport through $\dot{m}$ and $G$, consideration of the thermomechanical feedbacks along an idealized transect demonstrates the basic mechanisms through which ice infiltration limits bed strength. For a planar ice–till interface with $N < p_f$ everywhere, the local frictional resistance to sliding is $\tau = \mu N$ (consistent with the absence of a correlation between $\tau$ and $u$ on many Greenland outlet glaciers[4]). With amplified heterogeneities so that $N > p_f$ in some regions, the enhanced cohesion that accompanies ice infiltration causes the slip surface to migrate to the base of these zones; hence, the still liquid-saturated sediments beneath regions where ice cannot infiltrate as deeply represent topographic obstacles (see Fig. 4). Horizontal motion at the boundary between a debris-rich, ice-infiltrated keel and sufficiently weak liquid-saturated sediments can be accommodated by ploughing[20]. The resistance remains frictional while this occurs, but compaction of the ploughed sediments can lower $N$ and limit departures from $p_f$, thereby reducing ice infiltration and causing weakening with increased sliding velocity $u$[21]. If their separation $\lambda$ is sufficiently large and $u$ is sufficiently small, ploughing will cease once the frictional strength of obstacles exceeds the drag associated with deformation past them; this drag can be estimated as $[\tau] \sim 2C([h]/[\lambda])([u][h]/[A][\lambda]^2)^{1/n}$, where the factor $C \approx 30$ depends on geometrical details[22,23]. This transition is conceptually akin to a transition from form-to-skin-dominated drag in aerodynamics. For a given driving stress, the sliding velocity $u$ decreases with increasing $h$ in this regime, and in cases where the slowdown also

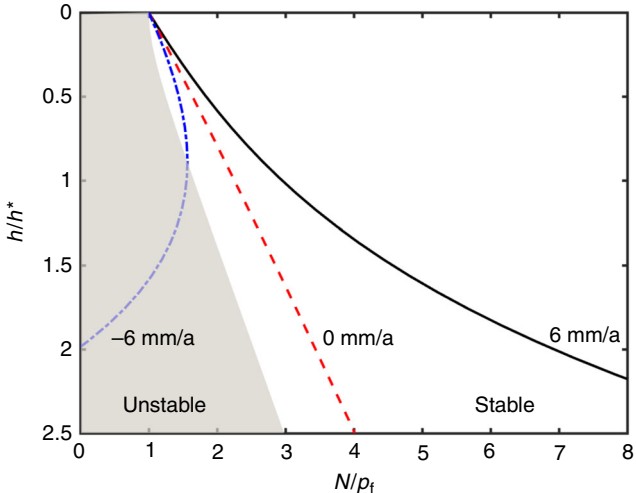

**Fig. 3 Freeze-on thickness.** Scaled steady-state thickness of ice-infiltrated sediments $h/h^\star$ (increasing downwards), plotted as a function of scaled effective stress $N/p_f$. For the nominal parameter values summarized in the Methods $h^\star \approx 1$ m and $p_f \approx 35$ kPa. The three curves, from left to right, are shown for cases with net freezing of 6 mm/a (blue dot dashed), balanced heat flux (red dashed), and net melting at 6 mm/a (solid black). The shaded region on the left indicates conditions during net freezing, where steady states are unstable to infinitesimal perturbations and instead transient ice infiltration can extend to much greater depths. A version of this plot contouring melt rate is provided as Supplementary Fig. 8

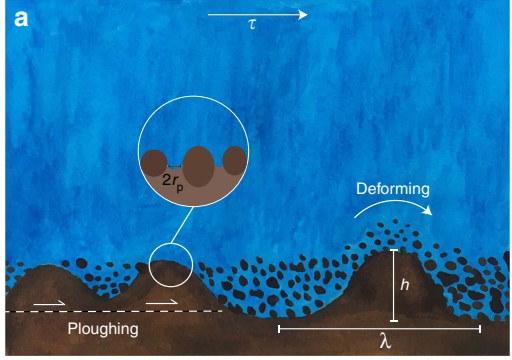

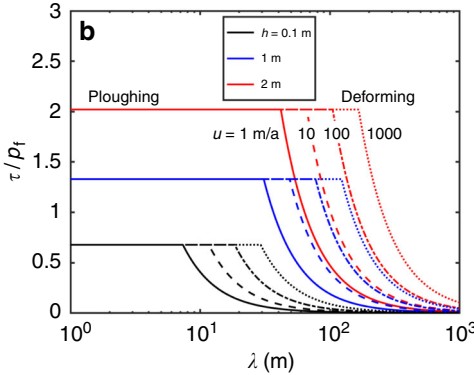

**Fig. 4 Sliding over beds roughened by heterogeneous freeze-on. a** Schematic illustration of deformation mechanisms that accommodate motion over basal sediments with heterogeneous ice infiltration of amplitude $h$ and wavelength $\lambda$. **b** The scaled shear stress $\tau/p_f$ as a function of wavelength $\lambda$ for obstacles with the amplitudes $h$ listed in the legend (with $\dot{m} = 0$ and values of $N/p_f$ from Fig. 3 of 1.1, 2.2, and 3.4, respectively). Cases with different sliding velocities are shown as solid lines for $u = 1$ m/a, dashed lines for $u = 10$ m/a, dot-dashed for $u = 100$ m/a, and dotted for $u = 1$ km/a. Ploughing of liquid-saturated sediments is favored for shorter wavelengths and lower amplitudes, whereas deformation is expected over stronger obstacles

decreases dissipation $\tau u$, the associated reduction in melt rate $\dot{m}$ can produce further increases in $h$. Ultimately, $h$ can become sufficiently large to anchor the overlying glacier and halt basal sliding so that internal deformation within the ice column becomes the dominant control on the basal shear stress, which should then follow the scaling given by Eq. (1).

Within the sliding regime, the resistance to slip posed by obstacles separated by smaller wavelengths is overcome by ploughing, whereas localized deformation more easily surmounts obstacles with smaller amplitudes. Figure 4b suggests that under glaciologically relevant conditions, the transition between these two mechanisms of accommodating slip takes place at wavelengths of a few tens of meters (see Supplementary Notes 3 and Supplementary Fig. 6). Balancing the drag imparted by deformation over obstacles with the gravitational driving force while implementing the mass balance condition as before gives

$$
\begin{aligned}
[\tau] &\sim \frac{[h]}{[\lambda]}\left(\frac{[2C]^n[a][l][\rho_i g][\sin\alpha]}{[A][\lambda]}\right)^{1/(n+1)} \\
&\sim [p_f]\left\{\frac{T_m}{\rho_i L[G][\lambda]}\left(\frac{[2C]^n[a][l][\rho_i g][\sin\alpha]}{[A][\lambda]}\right)^{1/(n+1)}\right\},
\end{aligned}
\tag{2}
$$

where the second expression is valid for $[h] \sim [h^\star]$. Setting $[\lambda] \geq 30$ m as an approximate lower limit and taking nominal values of $n = 3$, and $[A] = 2.4 \times 10^{-24}$ Pa$^{-3}$/s (ice at the melting temperature[24], $T = 273.15\,°C$), with $[a][l][\sin\alpha] = 10^2$ m$^2$/a (see Supplementary Fig. 6), the scaling given by Eq. (2) predicts that $[\tau] \leq 10^2$ $[h]$ kPa/m $\approx 3[p_f]$, where the latter equality assumes $h \sim h^\star$ and $[G] = 0.03$ K/m. The sensitivity of the scaling in Eq. (2) to variations in accumulation, upstream length, slope, and ice softness is weak, so alternative glaciologically relevant parameter choices are not expected to change the predicted upper limit on $[\tau]$ significantly. We conclude that expected meter-scale infiltration depths $[h]$ and the mechanical constraints that limit expected maxima in $[p_f]$ to $O(10^2\,\text{kPa})$ produce values of $[\tau]$ within the observed range illustrated in Fig. 1.

## Discussion

Variations are observed in both the thicknesses of debris-rich basal layers and the till depths over which deformation is concentrated, but data compilations suggest that decimeter to meter ranges are typical[25,26]. The similarity to expected values of $[h^\star]$ further supports the viability of thermo-mechanical feedbacks stemming from ice infiltration as a control on sliding strength. For simplicity, and in recognition of the tendency for the most rapid glacial sliding to take place over unconsolidated sediments we have focused on simple cases in which bedrock protrusions are of limited consequence. During such "soft-bedded" sliding, we have argued that surface energy and pore size combine to control the scale of effective stress ($p_f$) beyond which obstacle growth increases the basal roughness enough to retard significant sliding. In the opposite limit where "hard-bedded" conditions prevail, in principle the scaling in the first part of Eq. (2) from the main paper might still be considered valid, with $[h]$ reinterpreted as fixed over comparatively long-time scales. The prospect arises that an upper limit on $[\tau]$ during such hard-bedded sliding results from the tendency of ice sheets to use the debris-liberated upstream as tools to sculpt their basal interfaces and thereby limit the roughness in these regions as well.

Taking a broader perspective, modern treatments of large-scale ice sheet motion commonly avoid specifying mechanistic sliding relationships and instead assign local basal slipperiness values obtained by inverting modern observations of ice geometry and motion to postulate ad hoc connections between inferred sliding velocity and inferred shear stress[27,28]. The degree to which such

connections are predictive of future behavior depends on the manner in which basal conditions evolve. For example, the sensitivity of shear stress to sliding speed has the potential to increase, where higher effective stress levels cause obstacles to grow, but it can also decrease where effective stress drops enough that ice infiltration no longer occurs. The scaling presented here gives a plausible explanation for why the range of driving stress remains so limited for sliding glaciers in general. The roles of basal topography and hydrology in modulating sliding resistance have long been recognized. Our results highlight the need to also account for the evolution of the basal interface that results from thermo-mechanical feedback between ice, melt, and sediments.

## Methods

**Computation of freeze-on thickness**. The thickness of ice-infiltrated sediments at the base of a glacier must adjust to satisfy the constraints imposed by conservation considerations for heat, mass, and momentum. Rempel[12] formulated the transient problem, which can be reduced to an idealized one-dimensional second-order partial differential equation for temperature (see Supplementary Note 4). Alongside the specified temperature $T_f = T_m - \Delta T$ and its gradient $G$ (or the heat flux) at the base of the ice-infiltrated zone, the vertical force balance and Stefan condition provide the additional two boundary conditions needed to determine the unknown thickness $h$ of the ice-infiltrated zone and the temperature $T_l$ at its top. Important sediment constitutive behavior include the porosity $\phi$, the variation in ice saturation $S_i$ and permeability $k$ with temperature, and the density difference $\Delta\rho_{till}$ between the sediment particles and water $\rho_w$ (for simplicity the 10% density difference between ice and liquid is neglected). Figure 3 shows steady-state thicknesses, which satisfy

$$
h \approx \frac{N - p_f - \phi\frac{\rho_i L}{T_m}\int_{T_f}^{T_l} S_i \, dT - \eta\dot{m}\int_{z_f}^{z_l}\frac{(1-\phi S_i)^2}{k}\,dz}{(\rho_i L/T_m)G + \Delta\rho_{till}(1-\phi)g}.
\tag{3}
$$

The terms on the denominator of Eq. (3) account for the amount by which the vertical pressure exerted across wetting films and the buoyancy of the overlying till change with infiltration thickness—here the temperature gradient through the ice-infiltrated sediments $G$ is approximated as a constant (0.03 K/m—as expected for a nominal thermal conductivity of 2 W/(m K) and net conductive heat flux of 60 mW/m$^2$). The first term in the numerator is the effective stress $N$ at the base of the ice-infiltrated layer, which is equivalent to the load supported by the underlying sediment contacts. The second term, $p_f = \rho_i L(1 - T_f/T_m)$ (with temperatures in Kelvin), can be thought of as the load supported by the wetting films at the base of the ice-infiltrated layer. The third term is a small correction to the first term in the denominator, which accounts for the load supported by wetting interactions throughout the ice-infiltrated sediments. The final term in the numerator, involving the liquid viscosity $\eta$ and melt rate $\dot{m}$ accounts for deviations of the liquid pressure gradient away from hydrostatic equilibrium, which are assumed to follow Darcy's law. Sediment properties used to generate Fig. 3 are assigned so that $\phi = 0.35$, $\Delta\rho_{till} = 1650$ kg/m$^3$, and $T_m - T_f = 0.031$ K, with the ice saturation and permeability modeled using the power laws

$$
S_i = 1 - \left(\frac{T_m - T_f}{T_m - T}\right)^\beta \quad \text{and} \quad k = k_0\left(\frac{T_m - T_f}{T_m - T}\right)^\alpha,
\tag{4}
$$

with $\beta = 0.53$, $\alpha = 3.1$, and $k_0 = 4.1 \times 10^{-17}$ m$^2$ (see Table 2 in Rempel[29]). Together, these parameter choices yield $p_f \approx 35$ kPa and $h^\star = (T_m - T_f)/G \approx 1$ m.

**Data availability**. All of the data used in this paper are publicly available.

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

## Acknowledgements

We thank Bernard Hallet, Brad Lipovsky, Brent Minchew, Joe MacGregor, and Richard Hindmarsh for discussions and acknowledge financial support from NSF-1603907.

## Author contributions

A.W.R. and C.R.M. developed the primary arguments. A.S.D. compiled much of the data and performed the G.I.S. analysis related to Laurentide sliding. All authors contributed to the writing and editing.

## Additional information

**Competing interests:** The authors declare no competing interests.

