## [Peer Review File · Nature Communications]

Reviewers' comments:

Reviewer #1 (Remarks to the Author):

The manuscript presents a new theoretical concept aimed at explaining why basal shear stress of glaciers and ice sheets tend to be around 70 kPa or so. It also proposes a new and novel mechanism limiting the range of basal shear stresses.

I enjoyed reading the manuscript and I consider it is an excellent piece of work that deserves to be published.

There is a huge amount of carefully argued ideas presented in the paper and I expect the paper to be of great interest to the glaciological and the wider earth sciences community.

The authors convincingly argue that the limit on basal stress is not primarily related to ice rheology, but must for sliding glaciers be due to limits on bed strength. They provide a detailed examination of physical conditions at the ice-till interface and argue that in the case of unconsolidated sediments the basal stress is roughly equal to the effective stress N and furthermore that for clear ice sliding can only take place if $N < p_f$ where p_f is the pressure difference between ice and water within sediments. For $N > p_f$ the ice infiltrates into the underlying sediments, and the ice-infiltration distance, h , increases with N . This gives rise to ice-infiltrated keels and causes sliding velocity to decrease with increasing h , and ultimately this process may anchor the ice. In this picture, ice-infiltration acts as a control on sliding.

I found the analysis quite fascinating and the work appears to me to be a major theoretical breakthrough that may lead to reformulation of the basal boundary conditions used currently in large-scale ice-flow models.

I therefore strongly recommend the publication of this work.

Reviewer #2 (Remarks to the Author):

This paper aims to examine controls on the bed strength under sliding ice sheets and glaciers. Since sliding ice dominates the dynamically active regions which contribute to rapid change and sea level rise, understanding the bed properties under sliding ice is essential for building models which can accurately project future ice sheet behavior, particularly the transition between frozen-and-deforming ice and sliding ice. This makes it a very relevant topic for a journal such as Nature Communications. The authors claim that mechanical resistance posed by heterogeneous infiltration of ice into subglacial sediments (itself governed by local thermodynamics) is what governs the bed strength in sliding regions, and synthesize a theoretical underpinning to support that idea.

The paper is written carefully and well, and is bolstered by the supplementary material. The introduction, in particular, does a very good job of presenting the case for why understanding sliding and its related bed properties are important. I would support publication after a few minor points are addressed.

Specific comments:

1. line 54: "... in comparison with median normal stresses of..." -- I'd suggest clarifying that the normal stresses referred to here are computed using eqn 1. At least, that's where I think they're coming from... This sentence is a bit unclear, in that sense. If you have them, a representative map (or maps) of these stresses as you've computed them might be a nice addition to the supplementary material.

2. line 134: It would be useful to explicitly explain why the curves in the shaded region are

unstable. I'm presuming it's due to the negative slopes of the curves, but it took me a bit of effort to figure that out.

3. Figure 3: I'd suggest adding more curves than just one freezing, one no-melt, and one melting curve -- Perhaps counter intuitively, I think that a family of curves would make it easier to see what's going on in this figure. Otherwise, the freezing and non-freezing lines look like limits, which isn't the case. It would also make it a lot easier to see why the shaded boundary of the unstable region is where it is (presumably because it's where the individual curves each start to curve leftward (if that's not the case, then I'm missing something important))

4. line 141: perhaps explicitly point out that when the glacier is anchored to the frozen bed, it transitions out of the sliding regime into the purely deforming regime. Remember that you're writing for a somewhat more general audience than a glaciology journal.

5. line 144: though -> through

6. line 156: to help a more-general audience here, it might help to point out the parallel here with the concepts of form drag and frictional drag in aerodynamics.

7. line 159: "... u decreases with h..." -- this is ambiguous; you should explicitly state that u decreases with *increasing* h... (unless I missed something here)

8. line 159: "slow down" -> "slowdown"

9. line 172: it would be nice if you stated what temperature this value of A corresponds to. Most in a more-general audience won't have a good feeling for values of A.

10. Figure 1: I'm a bit skeptical of the values you computed using bedmap2, since it's generally too smoothed out and there are large regions which aren't well-constrained by observations, although I do admit that it's convenient. Have you made any effort to cross-validate these results against more-detailed datasets? This is presumably less of an issue for Greenland, since BedMachine has hopefully resolved some of these sorts of issues there.

Supplementary material:

1. Figure 3. Figure 3A is really nice. It might be more useful than Figure 1

2. Figure 5a. What does the multivalued solution mean for the net-freezing case? I think it means that there is no deformation solution available, but you should clarify that.

3. line 8 on page 10 -- "which is in consistent" -> "which is consistent" or "which is in agreement with"...

Reviewer #3 (Remarks to the Author):

The manuscript "Limits on bed strength beneath sliding glaciers" uses scaling relationships based on several existing theories of sliding and ice/bed interactions to explain that the upper limit of the driving stress of most glaciers and ice sheets is below a relatively small value of 150 kPa. The first part of their argument is based on rebuking arguments by Nye (1952) and Weertman (1957) that this can be explained by the shear thinning rheology of ice. I have some reservations about the use of scaling arguments for this part of the manuscript (see technical details below).

The second part of their argument is based on combining previous results by Rempel (2008) with sliding laws developed in Weertman (1957)/ Kamp (1970) to derive an alternative scaling relationship based on the infiltration of ice into sediment. The theory developed here is neat, but it appears to be only applicable for sliding along soft beds. However, there are also so-called

hard-beds which obey different sliding laws. I do not see how their theory could be applied to these settings.

The research presented in this paper is solid, albeit not entirely novel in the details. Instead, there are several previous results linked together by (very technical) scaling arguments into one argument aimed at explaining the observed limitations of the driving stress of glaciers and ice sheets. The result is interesting, but the manuscript in its current form does not make it clear what the implications for future studies are and how these results could be of use.

GENERAL COMMENTS:

I find the conclusion/discussion of the results unsatisfactory, especially for a high-impact journal. What do the results imply for modelling of sliding glaciers? How would one account for "the evolution of the basal interface" in models? Should glaciologists use a different sliding law?

It is generally assumed that the effective pressure N is set by the subglacial drainage system. Can the authors rule out that constraints on the subglacial water system could set a similar maximum basal shear stress?

Equation (2), figure 4 and the accompanying text are really the heart of the manuscript, but in the current manuscript the build-up to this part is rather lengthy and relies on several very technical arguments. As a reader, I found it difficult to follow these arguments without reading a lot of secondary literature. Given the journal, I think the authors should consider some simplification. For example, is it really necessary to discuss how far the ice infiltrates into the sediment in the main part of the manuscript, or could that be moved to the supplementary material?

Figure 4b appears to be the key to understanding the limitations on the basal shear stress and hence the driving stress, but it wasn't clear to me whether τ/p_f is the same as $[\tau]/[p_f]$ or whether it is calculated differently.

TECHNICAL COMMENTS:

Use of scaling arguments: My understanding of scaling relationships is that those only give the order of magnitude, but not real values. For example, both " $1 \sim 2$ " and " $1 \sim 10$ " can be considered true statements. Consequently, figure 3 in the supplementary material shows that τ_{deform} and τ_{drive} are indeed of the same order of magnitude over a wide range of values. Therefore, I am sceptical that scaling relationships alone are sufficient to exclude the rheology of ice as reason for the limitations on τ_{drive} and τ_{basal} . From the available data, it seems to me that it should be possible to estimate actual values of the vertical shear stress or basal shear stress from inversion. This would also be more consistent with the driving stress calculations, which are not scaled, but are values calculated from observational data.

Equation (1) does not appear in the referenced work by Nye and Weertman, but is basically a "shallow ice" relationship and should be referenced as such.

Supplementary: Section 2 should also include a discussion of Greenland. Are the results identical or significantly different?

We wish to thank the referees for highlighting potentially confusing aspects of the paper and suggesting areas for improvement. Throughout, the original comments are in black text and our responses are colored in blue text.

Comments to Referee 1

The manuscript presents a new theoretical concept aimed at explaining why basal shear stress of glaciers and ice sheets tend to be around 70 kPa or so. It also proposes a new and novel mechanism limiting the range of basal shear stresses. I enjoyed reading the manuscript and I consider it is an excellent piece of work that deserves to be published.

There is a huge amount of carefully argued ideas presented in the paper and I expect the paper to be of great interest to the glaciological and the wider earth sciences community. The authors convincingly argue that the limit on basal stress is not primarily related to ice rheology, but must for sliding glaciers be due to limits on bed strength. They provide a detailed examination of physical conditions at the ice-till interface and argue that in the case of unconsolidated sediments the basal stress is roughly equal to the effective stress N and furthermore that for clear ice sliding can only take place if $N < p_f$ where p_f is the pressure difference between ice and water within sediments. For $N > p_f$ the ice infiltrates into the underlying sediments, and the ice-infiltration distance, h , increases with N . This gives rise to ice-infiltrated keels and causes sliding velocity to decrease with increasing h , and ultimately this process may anchor the ice. In this picture, ice-infiltration acts as a control on sliding. I found the analysis quite fascinating and the work appears to me to be a major theoretical breakthrough that may lead to reformulation of the basal boundary conditions used currently in large-scale ice-flow models. I therefore strongly recommend the publication of this work.

Thank you for the supportive review.

Comments to Referee 2

Many thanks to reviewer 2 for clear and constructive comments.

Summary

This paper aims to examine controls on the bed strength under sliding ice sheets and glaciers. Since sliding ice dominates the dynamically active regions which contribute to rapid change

and sea level rise, understanding the bed properties under sliding ice is essential for building models which can accurately project future ice sheet behavior, particularly the transition between frozen-and-deforming ice and sliding ice. This makes it a very relevant topic for a journal such as Nature Communications. The authors claim that mechanical resistance posed by heterogeneous infiltration of ice into subglacial sediments (itself governed by local thermodynamics) is what governs the bed strength in sliding regions, and synthesize a theoretical underpinning to support that idea.

The paper is written carefully and well, and is bolstered by the supplementary material. The introduction, in particular, does a very good job of presenting the case for why understanding sliding and its related bed properties are important. I would support publication after a few minor points are addressed.

Specific comments

1. line 54: "... in comparison with median normal stresses of..." – I'd suggest clarifying that the normal stresses referred to here are computed using eqn 1. At least, that's where I think they're coming from... This sentence is a bit unclear, in that sense. If you have them, a representative map (or maps) of these stresses as you've computed them might be a nice addition to the supplementary material.

The normal stresses were calculated as overburden ice pressure, i.e. $\rho_i g(z_{\text{surface}} - z_{\text{bed}})$. We added maps showing the driving stress for Greenland and Antarctica in the supplement and now include a statement in the main text.

2. line 134: It would be useful to explicitly explain why the curves in the shaded region are unstable. I'm presuming it's due to the negative slopes of the curves, but it took me a bit of effort to figure that out.

The shaded region is unstable because there is a maximum effective stress that the fringe is able to support in steady state. In other words, the fringe thickness first grows with increasing effective pressure and then beyond the maximum, a larger fringe supports a lower effective pressure. If the effective pressure is prescribed as a value that is larger than the maximum, the fringe will continue to grow transiently, attempting to support the load. The source of this limit is related to the distribution of fluid pressures needed to draw water through ice-clogged pore space to satisfy mass conservation at a particular freezing rate. To avoid distracting the reader from the essential thrust of this manuscript, we prefer to omit such details and refer the reader to the more comprehensive description given in Rempel (2008).

3. Figure 3: I'd suggest adding more curves than just one freezing, one no-melt, and one melting curve – Perhaps counter intuitively, I think that a family of curves would make it easier to see what's going on in this figure. Otherwise, the freezing and non-freezing lines look like limits, which isn't the case. It would also make it a lot easier to see why the shaded boundary of the unstable region is where it is (presumably because it's where the individual curves each start to curve leftward (if that's not the case, then I'm missing something important)).

We have added a statement in the caption emphasizing that these are illustrative values and we also include a contour plot in the supplement.

4. line 141: perhaps explicitly point out that when the glacier is anchored to the frozen bed, it transitions out of the sliding regime into the purely deforming regime. Remember that you're writing for a somewhat more general audience than a glaciology journal.

This is a fair point. We have added a statement to this effect.

5. line 144: though → through.

Good catch.

6. line 156: to help a more-general audience here, it might help to point out the parallel here with the concepts of form drag and frictional drag in aerodynamics.

This is certainly a reasonable suggestion. We added a statement that this a conceptually similar transition. We do worry that there may be confusion between the slow viscous flow of glacier ice and turbulent flow, but we hope that the added conceptual connection will aid the readers' comprehension.

7. line 159: "... u decreases with h ..." – this is ambiguous; you should explicitly state that u decreases with *increasing* h ... (unless I missed something here).

Good suggestion.

8. line 159: "slow down" → "slowdown."

Great idea.

9. line 172: it would be nice if you stated what temperature this value of A corresponds to. Most in a more-general audience won't have a good feeling for values of A .

Excellent suggestion — we added a parenthetical statement after the definition of the value.

10. Figure 1: I'm a bit skeptical of the values you computed using bedmap2, since it's generally too smoothed out and there are large regions which aren't well-constrained by observations, although I do admit that it's convenient. Have you made any effort to cross-validate these results against more-detailed datasets? This is presumably less of an issue for Greenland, since BedMachine has hopefully resolved some of these sorts of issues there.

It is true that BEDMAP2 may not be the most ideal dataset and we welcome newer data. However, the point of the paper is not to evaluate the quality of BEDMAP2 data but rather describe a mechanism by which the strength of basal sediments are limited by heterogeneous ice infiltration into the underlying till. We have conducted sensitivity tests using reported uncertainties in BEDMAP2 data to explore their effects on inferred τ , and the results did not appear to change the character of the observations that our analysis seeks to explain.

Supplementary material

1. Figure 3. Figure 3A is really nice. It might be more useful than Figure 1.

We are glad you like it. Figure 1 clearly shows that this idea is relevant to all glaciers. Figure S3A shows the breakdown for Antarctica (and now Greenland as well), which is

very important yet in a short-format paper there isn't enough space to show everything at the same time. Thus, we prefer the current arrangement of figures.

2. Figure 5a. What does the multivalued solution mean for the net-freezing case? I think it means that there is no deformation solution available, but you should clarify that.

The multivalued solution is the same as in Figure 3, which leads to the unstable development of fringe — see description about line 134 above and reference to Rempel (2008).

3. line 8 on page 10 – “which is in consistent” → “which is consistent” or “which is in agreement with”...

Great catch.

Comments to Referee 3

We appreciate reviewer 3's frank assessment and suggestions for improvement.

Summary

The manuscript “Limits on bed strength beneath sliding glaciers” uses scaling relationships based on several existing theories of sliding and ice/bed interactions to explain that the upper limit of the driving stress of most glaciers and ice sheets is below a relatively small value of 150 kPa. The first part of their argument is based on rebuking arguments by Nye (1952) and Weertman (1957) that this can be explained by the shear thinning rheology of ice. I have some reservations about the use of scaling arguments for this part of the manuscript (see technical details below).

The second part of their argument is based on combining previous results by Rempel (2008) with sliding laws developed in Weertman (1957) / Kamp (1970) to derive an alternative scaling relationship based on the infiltration of ice into sediment. The theory developed here is neat, but it appears to be only applicable for sliding along soft beds. However, there are also so-called hard-beds which obey different sliding laws. I do not see how their theory could be applied to these settings.

The research presented in this paper is solid, albeit not entirely novel in the details. Instead, there are several previous results linked together by (very technical) scaling arguments into one argument aimed at explaining the observed limitations of the driving stress of glaciers and ice sheets. The result is interesting, but the manuscript in its current form does not make it clear what the implications for future studies are and how these results could be of use.

The controls on bed strength are of central importance for improved sliding parametrizations and forecasting future sea level rise. This remains an active area of research and the current contribution provides guidance for these effects, as emphasized throughout the manuscript and particularly in the closing paragraph.

GENERAL COMMENTS

1. I find the conclusion/discussion of the results unsatisfactory, especially for a high-impact journal. What do the results imply for modelling of sliding glaciers? How would

one account for “the evolution of the basal interface” in models? Should glaciologists use a different sliding law?

This is an important point and is the basis for the discussion in the supplement section ‘The Effects of Heterogeneous Sediment Entrainment on Bed Strength’. In this section, we describe the methods by which we calculate the stress scalings and the transition between ploughing and deformation that forms the basis for the ‘sliding law’ that we use in the main text.

2. It is generally assumed that the effective pressure N is set by the subglacial drainage system. Can the authors rule out that constraints on the subglacial water system could set a similar maximum basal shear stress?

This recognition is central to our work as well — we do not specify the effective stress and we assume it is set by a subglacial drainage system. Nevertheless, if the effective stress becomes too large (i.e. near an R-channel) and exceeds the threshold set by pore throat curvature p_f , then a frozen fringe will grow. Thus, our results are consistent with the typical Glaciological understanding that increasing effective stress leads to slower surface velocities.

3. Equation (2), figure 4 and the accompanying text are really the heart of the manuscript, but in the current manuscript the build-up to this part is rather lengthy and relies on several very technical arguments. As a reader, I found it difficult to follow these arguments without reading a lot of secondary literature. Given the journal, I think the authors should consider some simplification. For example, is it really necessary to discuss how far the ice infiltrates into the sediment in the main part of the manuscript, or could that be moved to the supplementary material?

While we understand the referee’s concern, this is largely a stylistic issue. We prefer to keep the current organization because it highlights the physical processes that underpin our model. Indeed, one of the central points we make is that the physics of sliding is sensitive to interactions with a dynamic bed. Ice infiltration is at the heart of this behavior and cannot be ignored.

4. Figure 4b appears to be the key to understanding the limitations on the basal shear stress and hence the driving stress, but it wasn’t clear to me whether τ/p_f is the same as $[\tau]/[p_f]$ or whether it is calculated differently.

Great point: the difference between τ/p_f and $[\tau]/[p_f]$ is that the first is variable that depends on λ and the second is a scaling for the two different regimes (ploughing versus deforming). This difference is discussed in the supplement section ‘The Effects of Heterogeneous Sediment Entrainment on Bed Strength’.

TECHNICAL COMMENTS

1. Use of scaling arguments: My understanding of scaling relationships is that those only give the order of magnitude, but not real values. For example, both “ $1 \sim 2$ ” and “ $1 \sim 10$ ” can be considered true statements. Consequently, figure 3 in the supplementary material shows that τ_{deform} and τ_{drive} are indeed of the same order of magnitude over a wide range of values. Therefore, I am sceptical that scaling relationships alone are sufficient to exclude the rheology of ice as reason for the limitations on τ_{drive} and τ_{basal} .

From the available data, it seems to me that it should be possible to estimate actual values of the vertical shear stress or basal shear stress from inversion. This would also be more consistent with the driving stress calculations, which are not scaled, but are values calculated from observational data.

This is an important discussion as the main results of the paper are presented as scaling analyses. We disagree with the referee's claim that the $1 \sim 10$ is a true scaling relationship as they are separated by an order of magnitude. Rather than the phrasing used by the referee, our point is that the rheology of ice *alone* is insufficient to explain the wide range of values. This shouldn't be surprising as it is well-known that large swaths of the ice sheets are sliding.

2. Equation (1) does not appear in the referenced work by Nye and Weertman, but is basically a "shallow ice" relationship and should be referenced as such.

A version of Equation (1) does appear on page 86 of Nye (1953) and page 37 of Weertman (1957). The footnote on page 36 of Weertman (1957) gives the scaling argument in words.

3. Supplementary: Section 2 should also include a discussion of Greenland. Are the results identical or significantly different?

This is a reasonable request. Therefore, we added a description of basal shear stresses in Greenland to the supplement. As is the case for Antarctica, the stress required to deform ice at observed surface velocities differs substantially from observed driving stresses, implying that the Nye/Weertman scaling breaks down.

References

- J. F. Nye. The flow law of ice from measurements in glacier tunnels, laboratory experiments and the Jungfraufirn borehole experiment. *Proc. R. Soc. Lond. Ser. A* 219, pages 477–489, 1953. doi: 10.1098/rspa.1953.0161.
- A. W. Rempel. A theory for ice-till interactions and sediment entrainment beneath glaciers. *J. Geophys. Res.*, 113(F1), 2008. ISSN 2156-2202. doi: 10.1029/2007JF000870. F01013.
- J. Weertman. On the sliding of glaciers. *J. Glaciol.*, 3(21):33–38, 1957. doi: 10.3198/1957JoG3-21-33-38.

Reviewer #3 (Remarks to the Author):

The manuscript "Limits on bed strength beneath sliding glaciers" by Meyer et al is clearly of scientific value. In response to the reviewer comments, the authors could have made more of an effort to address comments aimed at making the manuscript accessible to a broader audience, instead of generally opting to leave the manuscript as it is. I think that would have improved the impact of this paper, as indicated in my initial review. That said, I don't have any technical reservations at this point.

Nitpicking: The discussion concerning hard-bed sliding should be added to the main paper. It is an important limitation of the analysis.

It would be helpful if the relevant sections of the supplementary material were referenced in the main text rather than just the supplementary material as a whole.

Herein the original comments are in black text and our responses are colored in blue text.

Comments to Referee 3

Summary

The manuscript “Limits on bed strength beneath sliding glaciers” by Meyer et al is clearly of scientific value. In response to the reviewer comments, the authors could have made more of an effort to address comments aimed at making the manuscript accessible to a broader audience, instead of generally opting to leave the manuscript as it is. I think that would have improved the impact of this paper, as indicated in my initial review. That said, I dont have any technical reservations at this point.

Nitpicking

1. The discussion concerning hard-bed sliding should be added to the main paper. It is an important limitation of the analysis.

We have added the discussion of hard-bed sliding to the main text.

2. It would be helpful if the relevant sections of the supplementary material were referenced in the main text rather than just the supplementary material as a whole.

Following the *Nature Communications* style for referencing supplementary material, the relevant sections of the supplement are now clearer.